# The Impact of Turkey and Syria Earthquakes on University Students: Posttraumatic Stress Disorder Symptoms, Meaning in Life, and Social Support

**DOI:** 10.3390/bs13070587

**Published:** 2023-07-13

**Authors:** Othman A. Alfuqaha, Uday M. Al-masarwah, Randa I. Farah, Jehad A. Yasin, Leen A. Alkuttob, Nour I. Muslieh, Mouath Hammouri, Afnan E. Jawabreh, Duaa A. Aladwan, Randah O. Barakat, Nida’a H. Alshubbak

**Affiliations:** 1Counseling and Mental Health Department, Faculty of Educational Sciences, The World Islamic Sciences & Education University W.I.S.E., Amman 11947, Jordan; odai.masarweh@wise.edu.jo (U.M.A.-m.); doaa.aladwan@wise.edu.jo (D.A.A.); randa.odeh@wise.edu.jo (R.O.B.); 2Nephrology Division, Internal Medicine Department, School of Medicine, The University of Jordan, Amman 11942, Jordan; randafarah.11941@gmail.com; 3School of Medicine, The University of Jordan, Amman 11942, Jordan; jehadamerjehadyasin@gmail.com (J.A.Y.); leenalkotob@gmail.com (L.A.A.); nidaaalshubbak@yahoo.com (N.H.A.); 4Prince Salma Bnt Abdullah Nursing College, Al al-Bayt University, Al.Mafraq 130040, Jordan; nour2476@aabu.edu.jo; 5Department of Nursing, Jordan University Hospital, The University of Jordan, Amman 11942, Jordan; hmouath@gmail.com; 6Department of Obstetrics and Gynecology, The University of Jordan, Amman 11942, Jordan; afnanemad70@gmail.com

**Keywords:** meaning in life, social support, PTSD, earthquake, students, Turkey, Syria

## Abstract

(1) Background: Earthquakes are natural disasters that often result in significant loss of life and property. The objective of this study is to explore the frequency of symptoms associated with posttraumatic stress disorder (PTSD), levels of meaning in life (ML), and perceived social support (SS) among university students in the aftermath of the earthquake that occurred in Turkey and Syria in 2023. (2) Methods: This study utilizes a cross-sectional correlation design to achieve its objectives among 603 university students from two public universities using an online survey (via Google Forms). The survey was launched one month after the earthquake in Turkey and Syria and concluded on 6 April 2023. (3) Results: The results indicate that a total of 158 university students, constituting 26.20% of the sample, reported extreme symptoms of PTSD. The results also indicate that 184 students (30.51), and 140 students (23.22%) reported low levels of ML and SS, respectively. Female students were significantly more vulnerable to experiencing PTSD symptoms, as well as difficulties in finding ML and SS. Finally, results revealed that students who were pursuing postgraduate studies had a greater likelihood of exhibiting symptoms indicative of PTSD. (4) Conclusions: It is recommended that universities provide support services and resources for students experiencing PTSD symptoms.

## 1. Introduction

Earthquakes are a type of natural disaster that results in significant loss of life and property. In March 2023, a collective death toll of over 60,000 people was reported in Turkey and Syria as a result of earthquakes [1]. Traumatic events can leave a lasting impact on the mental well-being of those who experienced them, as well as those around them. One of the major anxiety disorders as a result of the disaster is posttraumatic stress disorder (PTSD). When individuals are exposed to a traumatic event, such as the recent coronavirus 2019 (COVID-19) and earthquakes in Turkey and Syria 2023, they may develop a debilitating mental health condition. Previous studies have indicated that PTSD symptoms are prevalent among university students, with some studies reporting rates ranging from 14% to 28% [2,3]. The symptoms of PTSD can include intrusive thoughts or memories, avoidance of stimuli related to traumatic events, and a heightened sense of arousal. The effects of PTSD can be profound, negatively impacting an individual’s daily functioning and overall quality of life, and it is important to understand the frequency and associated factors in different populations.

Meaning in life (ML) is a concept that has been widely explored in the field of psychology, particularly well-being and mental health. The meaning-making theory suggests that as humans, we possess an innate need to derive meaning from our experiences and existence, as it can provide us with a sense of purpose, motivation, and fulfillment [4]. The study has indicated that a sense of meaning and purpose in life are positively associated with well-being, including life satisfaction [5], life support [6], and resilience [7], as well as negatively related to existential vacuum [8], depression, and anxiety [9]. Studying the role of ML after traumatic events, like the recent earthquakes in Turkey and Syria in 2023, can offer insights into the psychological processes that promote resilience and recovery among university students by focusing on the role of social support and ML. Thus, students with higher resilience demonstrate greater hardiness when facing crises. This could inform interventions to enhance resilience and promote posttraumatic growth in those affected by trauma. The ML scale can provide valuable information about students’ resilience during crises.

Social support (SS), which refers to the resources provided by others that can be used to cope with stressors, has been found to be protective against mental health issues [10]. Previous studies have shown that individuals with higher levels of ML may be more resilient in the face of trauma and better able to cope with stress [11]. SS can provide individuals with emotional, informational, and tangible resources that can help them cope with stressors and may act as an important factor against the development of PTSD. The SS scale is a useful tool for investigating the importance of social and family networks during crises, offering insights into their role in promoting resilience and coping strategies.

University students are a critical population for examining the prevalence of PTSD symptoms, levels of ML, and perceived SS. Students are vulnerable to various stressors, such as academic demands, financial pressures, and social adjustment challenges, which may increase their risk of developing mental health difficulties, such as PTSD [12,13]. Additionally, recent traumatic events, such as earthquakes in the surrounding region of Turkey and Syria in 2023, may exacerbate this risk. The university setting offers a distinct chance to explore how the connections between having a sense of ML, SS, and PTSD symptoms interact with each other during a particular phase of an individual’s development. Understanding the experiences and needs of university students may aid in the identification of effective interventions to promote resilience and prevent or treat PTSD symptoms in this population. 

As far as we know, there have been no previous explorations into the frequency of PTSD symptoms, the extent of ML, and the perceived SS levels among university students following the earthquake that transpired in Turkey and Syria in 2023. Thus, this study seeks to contribute to the existing literature by examining the frequencies of PTSD symptoms, levels of ML, and perceived SS among university students in the aftermath of the earthquake that occurred in Turkey and Syria in 2023. Additionally, it examines the potential variations in these variables between male and female university students. The investigation will also examine how various sociodemographic factors, such as gender, age, family income, and educational levels, impact the development of PTSD.

## 2. Materials and Methods

### 2.1. Study Design

To ensure a rigorous and transparent account of the study methodology, results, and conclusions, this study utilizes a cross-sectional correlation design to achieve its objectives. 

### 2.2. Participants and Setting

For this study, participants were selected from two publicly funded universities situated in Amman, which is the capital city of Jordan. The universities were selected through a convenience sampling method, based on their accessibility to the students. To qualify for participation in this study, individuals had to be presently enrolled as students in one of the public universities, with any specialization, hold a bachelor’s or postgraduate degree, and express willingness to participate. Conversely, exclusion criteria comprised private university students or those who declined to participate. Private universities were excluded due to an extended approval process. To address the nonresponse issue, we conducted interviews with students who declined participation in the study. The students revealed that a common reason for their refusal was time constraints. Ethical considerations were upheld as we sought official permission from the selected universities to administer the online survey (via Google Forms) through official email and WhatsApp groups.

The online survey was launched one month after the occurrence of the earthquake in Turkey and Syria and concluded on 6 April 2023; the goal of the survey was to establish the exact frequency rate of PTSD according to the literature [14]. To obtain a representative study sample, the researchers determined that a total of 350 participants would be sufficient, based on a previous study conducted by Hair et al. [15]. A group of 603 students were included in the study and consented to participate, with one university contributing 340 valid responses and the other university contributing 263 valid responses. The study began with a cover letter that explained the purpose and procedures, followed by a question regarding the participants’ agreement to participate. The survey comprised four sections: (1) demographic factors—gender, age, family income, and educational levels; (2) Davidson Trauma Scale; (3) ML questionnaire; and (4) perceived SS scale. The order of the questionnaires was counterbalanced to avoid order effects. The study assistants’ phone numbers and emails were provided in case assistance was required. Before the study commenced, ethical clearance was granted by the university’s ethics committee, and participants gave their informed consent by clicking on the “I agree to participate” button before their involvement in the study.

### 2.3. Study Tools

#### 2.3.1. Davidson Trauma Scale (DTS)

It is a self-administered questionnaire designed to assess the presence and severity of posttraumatic stress symptoms in individuals who have experienced trauma. The DTS was developed by Davidson et al. [16], while the Arabic adaptation of the DTS underwent a rigorous evaluation process to determine its validity in the Arabic context. Various aspects, such as face validity, content validity, and construct validity, were thoroughly examined to ensure the reliability and appropriateness of the Arabic version. The outcomes of these evaluations are currently being considered for publication. The scale comprises 17 items, which are categorized into 3 dimensions—re-experiencing, hyperarousal, and avoidance—with each dimension including 5, 5, and 7 items, respectively. The Arabic version of the DTS demonstrated a high degree of internal consistency, with a calculated Cronbach alpha coefficient of 0.94. Participants rated each item on a 5-point Likert-type scale, ranging from 0 (Not at All Distressing) to 4 (Extremely Distressing). Scoring higher on the DTS suggests a greater presence of symptoms associated with PTSD.

#### 2.3.2. ML Questionnaire (MLQ)

It is a 10-item questionnaire that was designed by Steger and Frazier [17] to assess two dimensions (5 items each) of ML among university students: the presence of meaning, which gauges the extent to which individuals feel that their lives hold significance and purpose, and the search for meaning, which measures the extent to which individuals seek meaning in their lives. The MLQ has been translated into Arabic for the purposes of this study. The Arabic version of the MLQ exhibited robust psychometric properties, demonstrating a high-reliability coefficient of 0.91. Additionally, the construct validity of the Arabic MLQ was found to be satisfactory, as evidenced by intracorrelation coefficients exceeding 0.40 and a favorable Kaiser-Meyer-Olkin measure of 0.92. Each item was rated on a 7-point Likert-type scale ranging from 1 (Absolutely True) to 7 (Absolutely Untrue), with higher scores indicating a greater sense of ML.

#### 2.3.3. Multidimensional Scale of Perceived SS (MSPSS)

It is a 12-item questionnaire used to assess perceived SS among university students. The MSPSS was originally developed by Zimet et al. [18] and consists of three dimensions—significant other, family, and friends—with each consisting of four items. We utilized the Arabic version of the MSPSS, as translated by Ebrahim and Alothman [19]. The reliability of the MSPSS was evaluated using Cronbach’s alpha coefficient, which yielded a score of 0.95, indicating high internal consistency and reliability. Respondents rated each item on a 7-point Likert-type scale ranging from 1 (Very Strongly Disagree) to 7 (Very Strongly Agree).

### 2.4. Statistical Analysis

To summarize the demographic characteristics of the sample and the distribution of selected variables, descriptive statistics were utilized. The study also employed several statistical analyses to explore the associations between the variables of interest. The Pearson correlation coefficient (r) was used to examine the relationships between the selected variables, while binomial regression analysis was used to identify demographic factors associated with PTSD. Additionally, independent samples *t*-test and one-way ANOVA were conducted to investigate the differences in selected variables between demographic groups. The significance level for all statistical analyses was set at *p* < 0.05.

### 2.5. Ethical Consideration

The study was conducted with strict adherence to ethical principles and guidelines, and all participants provided informed consent. Ethical approval from the Research Ethics Committee of the Faculty of Educational Sciences at The World Islamic Sciences & Education University (Approval No: 5/1/9/405) was obtained prior to the study’s commencement.

## 3. Results

### 3.1. Demographic Characteristics

Within Table 1, we present an overview of the demographic characteristics of the study participants. It is evident from the table that a significant proportion of the sample consisted of female students currently enrolled in bachelor’s degree programs. Surprisingly, more than half of the participants were 21 years old or younger and their families earned less than JOD 350, which is approximately equal to USD 490 per month. The data were found to be normally distributed based on the skewness and kurtosis results. After asking participants about their religious beliefs (“Do you believe in a religion?”), it was found that 97.7% of them believed in God. Finally, more than one-third (37.7%) of the participants believed that an earthquake that happened in Turkey and Syria was a punishment from God.

### 3.2. Investigating the Prevalence and Gender Differences in PTSD, ML, and SS

The findings from Table 2 indicate that a total of 158 university students, constituting 26.20% of the sample, reported extreme symptoms of PTSD, with re-experiencing symptoms exhibiting the highest mean score. Additionally, the results reveal a statistically significant difference in the prevalence of PTSD symptoms between female and male students, with female students presenting higher levels of PTSD symptoms than their male counterparts. Regarding ML, the results indicate that 184 students (30.51%) reported low levels of meaning, with the presence of meaning being more commonly reported than the search for meaning. There was a significant difference found in the levels of ML between male and female students (*p* = 0.03), with female students reporting higher levels of ML than male students. Furthermore, 140 students (23.22%) reported low levels of SS, with the “friends” dimension showing the lowest mean score compared with other dimensions. The results reveal that female students had statistically significantly higher levels of SS than male university students (*p* = 0.05).

### 3.3. Association between the Selected Variables

The study used Pearson correlation coefficients to investigate the relationships between the variables under examination. The findings show that there was a significant positive correlation between PTSD symptoms and both ML and SS. Specifically, a positive association was found between PTSD symptoms and ML (*r* = 0.11, *p* = 0.009), as well as between PTSD symptoms and SS (*r* = 0.12, *p* = 0.003). This suggests that university students with more severe PTSD symptoms may require more SS and may be searching for greater meaning in their lives. Furthermore, ML was strongly and positively associated with SS (*r* = 0.63, *p* < 0.001), indicating that university students who perceive their lives as meaningful are more likely to report greater levels of SS

### 3.4. Factors Associated with PTSD

A regression analysis was performed to examine the relationship between PTSD and demographic factors. The findings are presented in Table 3. 

The results indicate that female university students were approximately 1.97 times more likely to exhibit PTSD symptoms than male students (AOR = 1.97, 95% CI = 1.20–3.22, *p* = 0.007). Additionally, postgraduate students had approximately 3.07 times greater odds of exhibiting PTSD symptoms than students with a bachelor’s degree (AOR = 3.07, 95% CI = 1.16–8.09, *p* = 0.02). However, the analysis did not reveal any significant association between PTSD and other demographic factors. 

## 4. Discussion

This study presents information on the prevalence of PTSD following the earthquake in Turkey and Syria. Additionally, it sheds light on potential interventions aimed at improving the overall mental health of university students who may be at risk of developing PTSD and experiencing low levels of ML and SS. However, the results reveal that a considerable proportion of students (26.20%) reported extreme symptoms of PTSD, with re-experiencing symptoms being the most commonly reported. In a previous study, it was reported that the occurrence of PTSD among French college students was 19.5% during the COVID-19 pandemic. However, our study reported a significantly higher prevalence [20]. Notably, it was found that female university students reported significantly more severe PTSD symptoms following the earthquake in Turkey and Syria, compared with their male counterparts. This finding highlights the potential gender-specific risk factors that may increase vulnerability to PTSD symptoms in the aftermath of natural disasters. This result is in line with a previous study that has emphasized the increased occurrence of PTSD symptoms in university students [21,22].

Around 30% of the study participants reported low levels of ML, with the majority of them indicating the presence of meaning rather than the search for meaning. Furthermore, our results suggest a higher prevalence of low ML among university students following the earthquake in Turkey and Syria compared with a previous study conducted during the post-epidemic period in China [23]. Following the earthquake, many university students have experienced difficulties in finding a sense of purpose and direction in their lives and a sense of existential meaninglessness. Our findings reveal an interesting gender difference, with female students reporting higher levels of ML after the earthquake compared with male students. These results suggest that female students may place greater importance on finding meaning in their lives than their male counterparts. In China and during the COVID-19 pandemic, female college students experienced a positive change in their perception of life meaning [24]. 

Approximately 23% of the students reported lower levels of SS, with the lowest dimension being supported by friends. This result is congruent with previous studies [25,26]. This emphasizes the significance of addressing SS during crises by implementing psychological intervention programs that aim to enhance SS from family, friends, and the community. We found that female students reported higher levels of SS than their male counterparts following the earthquake, which emphasizes the influence of gender on SS. This result is consistent with findings from previous studies [27,28]. 

The study’s results suggest that university students experiencing PTSD symptoms following an earthquake may have a greater need for ML and SS. Specifically, a positive association was found between PTSD symptoms and both ML and SS, implying that those with more severe PTSD symptoms may require more understanding in their lives, as well as more SS. During periods of extremely intrusive thoughts and re-experiencing traumatic events, university students may seek to find ML through prayer and by strengthening their connections with their families. Numerous studies have established a strong association between PTSD and both ML and SS [29,30]. Furthermore, the result findings reveal that there was a significant positive association between ML and SS. This implies that those who perceive their lives as meaningful are more likely to report higher levels of SS. These results underscore the need for targeted interventions that cater to the unique needs and experiences of female and male university students to enhance their overall lifestyles. A study conducted on 936 adolescents in China showed that perceived SS was associated with both ML and self-efficacy [27]. In contrast, a recent study found that PTSD symptoms were negatively associated with perceived SS [31]. 

The results of the study indicate that there is a significant correlation between PTSD symptoms and both gender and educational level among university students affected by the earthquake in Turkey and Syria in 2023. Specifically, the analysis indicates that female students and those with postgraduate education are at a higher risk of PTSD symptoms contrasted with their male and undergraduate counterparts. This could be related to their emotions, behaviors, and cognitive process. These outcomes are in line with a previous study that has shown a higher occurrence of PTSD symptoms among female university students relative to their male peers in response to traumatic events [32,33]. 

Postgraduate students are more likely to be married, so they have additional responsibilities of providing for their family’s needs, including food and safety. In contrast, a recent study conducted in Jordan found that age was the primary predictor of panic-buying behavior among the population [34]. These findings suggest that targeted interventions aimed at addressing PTSD symptoms among university students should consider gender and educational level as important factors. Moreover, it may be advantageous to prioritize interventions that foster a perception of meaning and SS among university students, particularly those who are female or unemployed, in order to promote their overall welfare.

Surprisingly enough, age, educational level, and family income were not associated with PTSD symptoms. The absence of an association between age and PTSD symptoms in our study could be attributed to the relatively narrow age range of the university student participants. As they were within a similar age group, the potential variations in life experiences and susceptibility to PTSD symptoms might not have been prominent. This finding contrasts with a previous study that suggested a higher vulnerability to mental health issues among young individuals [35].

The absence of a significant association between family income and PTSD symptoms suggests that the influence of financial resources alone on the development of PTSD may be limited. This finding implies that during crises, the availability of financial support may not necessarily serve as an effective form of social support for individuals experiencing PTSD symptoms. A recent investigation revealed that the experience of alienation affects individuals’ willingness to learn across diverse educational levels [36].

Although this study provides valuable insights into the prevalence and associations between PTSD, ML, SS, and demographic factors, it is not without limitations. First, it is important to note that the study was conducted solely in public universities; thus, the generalizability of the findings to other private university settings may be limited. Second, the study used a cross-sectional design, which limits the ability to draw causal inferences or examine changes over time. Third, one notable limitation of our study is the relatively high percentage of female participants (76%), which introduces a potential influence on the results. Finally, the study did not consider other potential confounding factors that may influence the relationships between the variables, such as trauma exposure history or mental health treatment history.

## 5. Conclusions and Recommendations

This study highlights the significant impact of earthquakes on university students’ mental health, particularly in terms of PTSD symptoms. The findings highlight a high prevalence of PTSD symptoms. It suggests that female students may be particularly vulnerable to experiencing PTSD symptoms, as well as experiencing difficulties in finding ML and SS. Moreover, positive associations were noted between PTSD, ML, and SS. Finally, being a postgraduate student was associated with an increased likelihood of having PTSD symptoms. Based on the findings, it is recommended that universities provide support services and resources for students experiencing PTSD symptoms and low levels of ML and SS. Specifically, interventions aimed at improving well-being should focus on addressing the complex interplay between PTSD symptoms, ML, and SS. Tailored interventions should be considered by universities to meet the specific needs of female students, based on the gender differences identified. Finally, future studies should investigate the long-term consequences of such traumatic events, including their potential impact on chronic health conditions.

## Figures and Tables

**Table 1 behavsci-13-00587-t001:** Demographic Characteristics (N = 603).

Factor	Description	Frequency (%)	Skewness	Kurtosis
Gender	Male	144 (23.9)	1.23	0.49
	Female	459 (76.1)		
Age (Yrs.)	≤21	358 (59.4)	0.38	1.8
	>21	245 (40.6)		
Family income per month (JOD)	<350	306 (50.7)	0.83	0.42
	350–749	165 (27.4)		
	750–3000	111 (18.4)		
	>3000	21 (3.5)		
Educational level	Bachelor	569 (94.4)	1.84	1.92
	Postgraduate	34 (5.6)		

Note. Yrs.: years; JOD 1.00 is approximately USD 0.70; %: percentage.

**Table 2 behavsci-13-00587-t002:** Measurement Levels and Gender Differences (N = 603).

Variables	M	SD	Min	Max	Skewness	Kurtosis
Re-experiencing (5 items)	5.22	4.23	0	20	0.92	0.63
Avoidance (7 items)	7.11	5.50	0	28	0.90	0.96
Hyperarousal (5 items)	3.85	4.40	0	20	1.31	1.21
Total PTSD	16.18	12.74	0	68	1.04	1.27
Gender			Frequency	Percentage	*t*-test	*p*-value
Male	12.6	12.4	144	23.9	3.93	<0.001 **
Female	17.5	12.6	459	76.1		
Presence of meaning (5 items)	27.15	7.30	5	35	−0.89	−0.28
Search for meaning (5 items)	26.89	8.68	5	35	−1.04	−0.10
Total ML	54.04	14.37	16	70	−1.05	0.07
Gender			Frequency	Percentage	*t*-test	*p*-value
Male	51.8	15.7	144	23.9	2.20	0.03 *
Female	54.7	13.8	459	76.1		
Significant Other	21.41	7.53	4	28	−0.94	−0.45
Family	21.49	6.89	4	28	−1.01	−0.16
Friends	18.72	6.07	4	28	−0.60	−0.39
Total SS	61.63	18.77	12	84	−0.98	−0.08
Gender			Frequency	Percentage	*t*-test	*p*-value
Male	59.2	19.9	144	23.9	1.95	0.05 *
Female	63.5	18.3	459	76.1		

Note. M: mean. SD: standard deviation; Min: minimum; Max: maximum; *t*-test: independent sample *t*-test; * *p* < 0.05. ** *p* < 0.001.

**Table 3 behavsci-13-00587-t003:** Binomial Regression Analysis for PTSD with Demographic Factors (N = 603).

Variables	Adjusted OR (95% CI)	*p*-Value
Gender		
Male	1.00 (reference)	
Female	1.97 (1.20–3.22)	0.007 **
Age (Yrs.)		
≤21	1.00 (reference)	
>21	0.99 (0.68–1.45)	0.91
Educational Level		
Diploma degree	1.00 (reference)	
Bachelor’s degree	0.34 (0.02–5.05)	0.43
Postgraduate	0.38 (0.03–4.67)	0.45
Family income per month (JOD)		
<JOD 350	1.00 (reference)	
JOD 350–749	0.34 (0.07–1.52)	0.15
JOD 750–3000	0.38 (0.08–1.82)	0.22
>JOD 3000	0.26 (0.06–1.22)	0.08
Educational level		
Bachelor	1.00 (reference)	
Postgraduate	3.07 (1.16–8.09)	0.02 *

Note. Yrs.: years; 1 JOD: JOD 1.00 is approximately USD 0.70; *p*-value of the Wald chi-square statistics; * *p* < 0.05; ** *p* < 0.01; 95% CI = 95% confidence interval.

## Data Availability

The data presented in this study are available upon request from the corresponding author.

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
