# Peer review of "The Impact of Turkey and Syria Earthquakes on University Students: Posttraumatic Stress Disorder Symptoms, Meaning in Life, and Social Support"

_behavsci, 2023, doi:10.3390/bs13070587_

Round 1

Reviewer 1 Report

Timely and early examination of the impact of disaster on university level students. Paper could be improved by more details about the sample.  Including if available: characteristics of non-respondents. For example there should be some discussion of the possible reason that 76% of the sample are female and what this might  have meant for the findings of higher levels of distress in females.  Is this consistent with the literature. I think it should be discussed as a potential limitation. Is there any information on personal/family/ social group having direct experience of harmful impact. This could be useful in seeking government or other sources of aid.

I have not seen an academic paper that has noted religious convictions--this is potentially a strength and would like an explanation as to why this was not followed up in subsequent analyses.

In a paper written for  international readers the comment about the age and income level being "surprising" needs more information and more discussion.

The section on "demographic characteristics" needs minor editing. Are the educational levels given --completed--or current level of study the section relating to this is confusing.

In the introduction it is noted that the study can offer insights into "psychological process that promote resilience and recovery". I assume this is covered in the questionnaire or can be linked to the scales used but given the interest of this study to readers who may not be familiar with the scales used I think it would be useful to spell this out more explicitly.

Author Response

Response to Reviewers

Dear editor,

We would like to thank you and the reviewer for their valuable comments and feedbacks. Point-by-point responses to reviewers are listed below.

Reviewer #1:

Comment 1:

Timely and early examination of the impact of disaster on university level students. Paper could be improved by more details about the sample.  Including if available: characteristics of non-respondents. For example there should be some discussion of the possible reason that 76% of the sample are female and what this might  have meant for the findings of higher levels of distress in females.  Is this consistent with the literature. I think it should be discussed as a potential limitation. Is there any information on personal/family/ social group having direct experience of harmful impact. This could be useful in seeking government or other sources of aid.

Response 1:

Thank you for your valuable feedback on our paper. Regarding your suggestion to provide more details about the sample, including characteristics of non-respondents, we agree that additional information would enhance the study's comprehensiveness. A new paragraph was added as follows line (99-102): “To address the non-response issue, we conducted interviews with students who declined participation in the study. The students revealed that a common reason for their refusal was time constraints.

Moreover, we added in the limitation section what you mentioned according to 76% of female participants. Thus, a new paragraph was added as follows: Thirdly, one notable limitation of our study is the relatively high percentage of female participants (76%), which introduces a potential influence on the results.

Furthermore, in our manuscript, we mentioned that our results are in line with previous study that has shown a higher occurrence of PTSD symptoms among female university students relative to their male peers in response to traumatic events [33,34].

Additionally, we acknowledge the significance of exploring whether participants or their personal/family/social groups had direct experiences of harmful impacts. This information would indeed be useful in identifying avenues for government or other sources of aid.

Comment 2:

I have not seen an academic paper that has noted religious convictions--this is potentially a strength and would like an explanation as to why this was not followed up in subsequent analyses.

Response 2:

We appreciate the reviewer's observation and suggestion regarding the inclusion of religious convictions in our study. While we agree that examining religious beliefs could be an important aspect to consider, we would like to clarify our rationale for not including it in subsequent analyses.

In this particular study, our focus was primarily on investigating the impact of the earthquakes on posttraumatic stress disorder symptoms, meaning in life, and social support among university students. While religious convictions could potentially be relevant and influence these outcomes, we made the decision to prioritize the specific variables outlined in the paper due to their established connections with post-disaster experiences and psychological well-being. We acknowledge that exploring the role of religious convictions in subsequent analyses could have provided valuable insights. However, given the scope and objectives of our study, we opted to delve deeper into other factors that were directly related to our research questions. Nonetheless, we recognize the significance of religious beliefs and their potential influence on post-disaster experiences, and we encourage further research to investigate this aspect in more detail. Thank you for bringing this point to our attention, and we will ensure to address the reviewer's feedback and consider the inclusion of religious convictions in future studies related to disaster impacts on university students.

Comment 3:

In a paper written for international readers the comment about the age and income level being "surprising" needs more information and more discussion.

Response 3:

As your suggestion, we added more discussion in our revised manuscript as follows: Surprisingly enough, age, educational level, and family income were not associated with PTSD symptoms. The absence of an association between age and PTSD symptoms in our study could be attributed to the relatively narrow age range of the university student participants. As they were within a similar age group, the potential variations in life experiences and susceptibility to PTSD symptoms might not have been prominent. The absence of a significant association between family income and PTSD symptoms suggests that the influence of financial resources alone on the development of PTSD may be limited. This finding implies that during crises, the availability of financial support may not necessarily serve as an effective form of social support for individuals experiencing PTSD symptoms.

Comment 4:

The authors should use STOBE checklist to check the completeness of their methods section.

Response 4: Thank you for your comment. We appreciate your suggestion regarding the use of the STOBE checklist to ensure the completeness of our methods section. We would like to clarify that we indeed employed the STOBE (Strengthening the Reporting of Observational Studies in Epidemiology) checklist during the preparation of our manuscript. However, due to the limitations of manuscript length and to enhance readability, we included an appendix containing the complete STOBE checklist as supplementary material during the submission process. This appendix provides a comprehensive overview of how we adhered to the STOBE guidelines in reporting our study methods.

Comment 5:

The section on "demographic characteristics" needs minor editing. Are the educational levels given --completed--or current level of study the section relating to this is confusing.

Response 5:

Thank you for your feedback. We apologize for any confusion caused by the section on "demographic characteristics" in our manuscript. To clarify, the educational levels mentioned in the table represent the current level of study being pursued by the participants. Specifically, the majority of the sample consisted of female students enrolled in bachelor's degree programs. We will make sure to revise the section to provide clearer information regarding the educational levels of the participants. However, a new paragraph was added as follows: Within Table 1, we present an overview of the demographic characteristics of the study participants. It is evident from the table that a significant proportion of the sample consisted of female students currently enrolled in bachelor's degree programs.

Comment 6:

In the introduction it is noted that the study can offer insights into "psychological process that promote resilience and recovery". I assume this is covered in the questionnaire or can be linked to the scales used but given the interest of this study to readers who may not be familiar with the scales used I think it would be useful to spell this out more explicitly.

Response 6:

Thank you for bringing up this concern. In response to the reviewers' suggestion, we clarified the connection between the questionnaire used and the psychological processes that promote resilience and recovery. Please see the new paragraph was added in our revised manuscript as follows:

“Studying the role of ML after traumatic events, like recent earthquakes in Turkey and Syria 2023, can offer insights into the psychological processes that promote resilience and recovery among university students by focusing on the role of social support and ML. Thus, students with higher resilience demonstrate greater hardiness when facing crises. This could inform interventions to enhance resilience and promote post-traumatic growth in those affected by trauma. The ML scale can provide valuable information about students' resilience during crises”.

“The SS scale is a useful tool for investigating the importance of social and family networks during crises, offering insights into their role in promoting resilience and coping strategies.”

We hope now that our revised manuscript is acceptable for publication.

Reviewer 2 Report

Lines (68-69) need references

Gender, age, family income & educational levels contribute to PTSD? Not clear

88-89 established global guidelines? not clear

104 "studyers"? not clear

conducted by [13] (105) not clear

Arabic versions of DTS, MLQ & MSPSS: how were psychometrics determined?

English language review is required; edits need to be made for clarity of statements.

Author Response

Response to Reviewers

Dear editor,

We would like to thank you and the reviewer for their valuable comments and feedbacks. Point-by-point responses to reviewers are listed below.

Reviewer #2:

Comments and Suggestions for Authors

Comment 1: Lines (68-69) need references.

Response 1: 2 references were added.

Comment 2: Gender, age, family income & educational levels contribute to PTSD? Not clear

Response 2: We paraphrased the sentence as follows to be clear “The investigation will also examine how various sociodemographic factors, such as gender, age, family income, and educational levels, impact the development of PTSD. Line (83-85).

Comment 3: 88-89 established global guidelines? not clear.

Response 3: The sentence was deleted.

Comment 4: 104 "studyers"? not clear

Response 4: We replace it with “researchers”.

Comment 5: conducted by [13] (105) not clear.

Response 5: We paraphrased the sentence as “conducted by Hair et al., [15].

Comment 6: Arabic versions of DTS, MLQ & MSPSS: how were psychometrics determined?

Response 6:

Arabic version of DTS: A new paragraph was added as follows: “The DTS was developed by Davidson et al. [16], while the Arabic adaptation of the DTS underwent a rigorous evaluation process to determine its validity in the Arabic context. Various aspects such as face validity, content validity, and construct validity were thoroughly examined to ensure the reliability and appropriateness of the Arabic version. The outcomes of these evaluations are currently being considered for publication”.

Arabic version of MLQ: A new paragraph was added as follows: The Arabic version of the MLQ exhibited robust psychometric properties, demonstrating a high reliability coefficient of 0.91. Additionally, the construct validity of the Arabic MLQ was found to be satisfactory, as evidenced by intra-correlation coefficients exceeding 0.40 and a favorable Kaiser-Meyer-Olkin measure of 0.92.

Arabic version of MSPSS: As we mentioned in our manuscript line (144-150), we utilized the Arabic version of the MSPSS, as translated by Ebrahim and Alothman [19]. The reliability of the MSPSS was evaluated using Cronbach's alpha coefficient, which yielded a score of 0.95, indicating high internal consistency and reliability. Thus, we took it as already translated in Arabic language and we found only the reliability as mentioned.

Comments on the Quality of English Language

English language review is required; edits need to be made for clarity of statements.

Response:

We express our gratitude to the reviewer for bringing attention to the clarity of statements in our manuscript. Taking their feedback into careful consideration, we conducted a comprehensive review of the manuscript to enhance its clarity for readers.

We hope now that our revised manuscript is acceptable for publication.
